# Leveraging Thermal Infrared Imaging for Pig Ear Detection Research: The TIRPigEar Dataset and Performances of Deep Learning Models

**DOI:** 10.3390/ani15010041

**Published:** 2024-12-27

**Authors:** Weihong Ma, Xingmeng Wang, Simon X. Yang, Lepeng Song, Qifeng Li

**Affiliations:** 1Information Technology Research Center, Beijing Academy of Agriculture and Forestry Sciences, Beijing 100097, China; mawh@nercita.org.cn; 2School of Electronic and Electrical Engineering, Chongqing University of Science & Technology, Chongqing 401331, China; 2022204009@cqust.edu.cn; 3Advanced Robotics and Intelligent Systems Laboratory, School of Engineering, University of Guelph, Guelph, ON N1G 2W1, Canada; syang@uoguelph.ca

**Keywords:** thermal infrared imaging, pig state monitoring, deep learning for object detection, precision livestock farming

## Abstract

This study developed a new dataset, TIRPigEar, consisting of 23,189 thermal infrared images of pig ears to help monitor pig state. Pig ears are suitable for thermal imaging due to their unique vascular structure, which can produce distinct thermal patterns. Although ear temperature is variable and influenced by factors such as thermoregulation, we can still obtain an average temperature value for the ear region through image-based detection methods, which provides useful insights into the state of pigs. Using an inspection robot, these images were collected in real pig farm environments, labeled, and annotated for use in deep learning models. This dataset offers a non-contact, efficient method for analyzing pig temperature, making it a valuable resource for improving state monitoring in livestock farming, supporting earlier intervention and better management (FLIR Tools software (version Flir Tools 201804), FLIR IP Config software (version FLIR_IP_Config_3_5)).

## 1. Introduction

Farm management in pig farms is crucial for modern livestock production, and infrared thermography offers a non-contact monitoring method for accurate swine management assessment [1]. By enabling real-time status monitoring, infrared thermography facilitates early interventions in response to abnormalities, thereby mitigating potential risks. However, publicly available datasets for infrared thermographic applications in swine state monitoring are limited, which hinders researchers in computer science from performing swine state analysis and applications based on open datasets.

Nie et al. [2] applied deep learning algorithms to recognize pig facial expressions to assess pigs’ emotions, health status, and intentions, achieving a mean Average Precision (mAP) of 89.4%. Pu and Liu [3] proposed an improved YOLOv5 network for pig detection, reaching 1.6% higher detection accuracy than the baseline model in detecting feeding pigs. Ma et al. [4] developed a lightweight pig face recognition method, demonstrating high detection accuracy on small datasets and the potential to generate large-scale unannotated pig face data. Wang et al. [5] introduced a method for detecting dropped ear tags in breeding pigs, with a detection accuracy of 90.02% and a speed of 25.33 frames per second (fps), achieving real-time, accurate monitoring. However, the datasets used in these studies were not publicly released.

Esper et al. [6] introduced a pig carcass cutting dataset comprising RGB images, depth data, and camera intrinsic matrices for meat processing automation applications. Wang et al. [7] developed a dataset containing 2300 annotated infrared images of group-housed pigs for instance segmentation, reaching a segmentation accuracy of 97.9%. Shao et al. [8] established a dataset of 3200 images covering four pig postures, achieving an application accuracy of 92.45%. Bergamini et al. [9] observed approximately six million pigs and provided annotations for various behavioral patterns, including five individual behaviors, forming a large dataset for pig detection, tracking, and behavior analysis. Most existing datasets primarily consist of RGB images; however, infrared thermography (IRT), a non-invasive temperature measurement technique, can eliminate stress caused by direct contact with thermometers [10]. There is currently no available dataset specifically for pig ear recognition in infrared thermography.

Infrared thermal imaging enables precise temperature detection. Fan et al. [11] utilized infrared thermography for high-throughput plant phenotyping. Gadhwal et al. [12] extracted the canopy temperature of maize from thermal images, with a measurement error under 2 °C. Dang et al. [13] used UAV-based infrared thermal imaging to monitor the temperature of drip-irrigated cotton, achieving a minimum temperature error of 2.78 °C. Zhang et al. [14] reconstructed a 3D chicken model from RGB–Depth–Thermal maps, offering enhanced feather damage assessment compared to 2D thermal infrared or color images, thereby providing a reference for poultry farming research. IRT imaging is also suitable for early disease detection, respiratory state monitoring, vaccination evaluation, temperature monitoring, and estrus detection in pigs [15]. However, high-quality publicly available infrared thermal image datasets for swine and accurate temperature measurement methods for pigs are still lacking.

Given these gaps, this study establishes an infrared thermal image dataset focused on pig ear recognition and temperature measurement. The processes of data collection, cleaning, and annotation for the TIRPigEar dataset are discussed in detail, as well as the dataset’s performance across different deep learning models. The primary contributions of this study include the following:Establishing and publicly releasing an infrared thermal imaging dataset of pig ears with temperature information.Validating the dataset’s effectiveness through training with state-of-the-art deep learning models.Providing a potential way for the research on pig ear temperature analysis and anomaly detection to facilitate early intervention based on this dataset.

## 2. Materials and Methods

This section describes in detail the data collection of gestating sows in individual stalls using a swine inspection robot and an infrared thermal imaging camera, as well as the training and evaluation of different object detection models.

The experiments were conducted at the Guanghui Core Breeding Farm (Mianyang, China), a flagship pig-breeding facility of the Tieqi Lishi Group, located in Santai County, Mianyang, Sichuan Province, China. Tieqi Lishi Group, a national leader in agricultural industrialization, produces three million high-quality piglets annually, making the Guanghui farm an ideal site for obtaining extensive swine samples. The study’s subjects were 150 gestating Large White sows kept in individual stalls with dimensions of 220 cm (length), 75 cm (width), and 106 cm (height). The inspection robot moved steadily between two rows of stalls at a speed of 0.1 m/s, capturing thermal images of pig ears with an infrared thermal camera to form the final dataset.

Data collection was performed using an inspection robot (Beijing, China), previously proven effective in pig farms (Figure 1). Designed and manufactured by the Information Technology Research Center of the Beijing Academy of Agriculture and Forestry Sciences (NERCITA, Beijing, China), this robot significantly reduces the cost of manual data collection while providing high operational stability [16]. During the robot’s autonomous patrol, the thermal imaging camera captured images at a frequency of one frame per second. The specifications for the inspection robot and the thermal imaging camera are presented in Table 1.

### 2.1. Experimental Method

Thermal infrared images were captured at various angles in the indoor environment of a pig farm using an inspection robot. Based on the robot’s movement angles and the pigs’ activities, three viewing angles were established for data collection, as illustrated in Figure 2. The robot’s lifting mechanism allowed the infrared thermal camera to move parallel to the longitudinal axis, maintaining a perpendicular distance of 60 cm from the sow stalls. During data collection, external factors such as ambient temperature, lighting, humidity, and target occlusion in the pig farm were considered to capture samples across diverse environmental conditions.

Ear image data were collected through the camera mounted on the inspection robot, then filtered to yield a high-quality thermal infrared image dataset. This dataset was subsequently used to train object detection models to develop different pig ear detection algorithms. Once trained, these detection models were deployed on the inspection robot to predict and automatically extract pig ear regions from real-time thermal images collected during inspection. The workflow for the inspection robot’s data collection and analysis process is shown in Figure 3.

Due to the robot’s dynamic movement, camera shake and distortion might occur. Therefore, after completing the thermal infrared image collection, we manually verified the images to exclude invalid images, such as those with fast-moving pigs or large occlusions. The final dataset, filtered for quality, provides reliable, valuable data for research. The entire collection, conducted from April to June 2023, resulted in 23,189 images. The annotated dataset file size is 11.6 GB, with a compressed size of 7.71 GB.

### 2.2. Experimental Algorithm

We conducted thermal infrared pig ear detection experiments using mainstream single-stage object detection networks. Specifically, the thermal infrared pig ear detection models varied in size according to network width and depth, and included models from the YOLOV5, YOLOV6, YOLOV7, YOLOV8, RT-DETR, YOLOV9, YOLOV10, and YOLOV11 series. Comparative parameters of these models are presented in Table 2.

YOLOv5 is a real-time object detection model released by Glen Jocher, founder of Ultralytics, in 2020. It is known for its balance between speed and accuracy, making it suitable for applications where real-time performance is crucial. The thermal infrared pig ear detection model based on YOLOv5 is shown in Figure A1 in Appendix A. YOLOv5 was selected for its established reliability in various real-time detection tasks.

YOLOv6 was released by Meituan in 2022 and is focused on industrial applications, offering improvements in efficiency and accuracy for tasks that involve complex and diverse object detection scenarios. The structure of the YOLOv6-based thermal infrared pig ear detection model is shown in Figure A2. This model was chosen for its optimization in industrial contexts, which aligns with the demands of our detection environment.

YOLOv7, released by Chien-Yao Wang et al. in 2022, prioritizes enhancing both speed and accuracy for real-time object detection tasks. Its superior performance in real-time applications made it a strong candidate for this study. The model structure based on YOLOv7 is shown in Figure A3. We selected YOLOv7 for its balanced performance in both precision and inference speed, which is essential for our detection requirements.

YOLOv8, introduced by Ultralytics in 2023, focuses on high inference speed, particularly for deployment on edge devices with limited resources. This version has demonstrated cutting-edge performance in terms of both image detection accuracy and speed. The structure of the thermal infrared pig ear detection model based on YOLOv8 is shown in Figure A4. YOLOv8 was selected for its exceptional performance in edge deployment, making it ideal for real-time applications where computational resources are constrained.

RT-DETR, released by Baidu in 2023, is a real-time object detection framework that became a state-of-the-art (SOTA) model at the time due to its speed and accuracy. RT-DETR allows for direct prediction of pig ears from thermal infrared images without the need for pre-defined anchor boxes or candidate boxes. This method reduces computational costs and helps minimize false positives and missed detections. The RT-DETR-based model structure is shown in Figure A5. We opted for RT-DETR for its efficiency and ability to streamline the detection process without sacrificing accuracy.

YOLOv9, released in 2024 by the same team behind YOLOv7 (Chien-Yao Wang et al.), focuses on addressing inherent information loss challenges in deep neural networks. The structure of the YOLOv9-based thermal infrared pig ear detection model is shown in Figure A6. YOLOv9 was chosen for its advancements in information retention, which enhances the model’s ability to detect features in challenging thermal infrared images.

YOLOv10, released in 2024 by the Tsinghua University research team, builds on the Ultralytics Python package and aims to address deficiencies in post-processing and model architecture for real-time object detection tasks. The structure of the YOLOv10-based thermal infrared pig ear detection model is shown in Figure A7. YOLOv10 was selected for its novel improvements in post-processing and architecture, providing better overall accuracy and efficiency for our specific task.

YOLOv11, the latest iteration in the Ultralytics YOLO series of real-time object detectors, redefines what’s possible in terms of accuracy, speed, and efficiency. It pushes the boundaries of state-of-the-art performance in detection tasks. The structure of the YOLOv11-based thermal infrared pig ear detection model is shown in Figure A8. YOLOv11 was chosen for its cutting-edge performance, offering the best combination of speed and precision for our study.

YOLOV5 utilizes the C3 module, which leverages Cross Stage Partial (CSP) architecture [17], and the SPPF module based on Spatial Pyramid Pooling (SPP) [18]. It also incorporates the FPN structure (Feature Pyramid Network [19]) and the PAN structure (Path Aggregation Network [20]). YOLOV6 includes RepVGG [21] and CSPStackRep, built on the CSP framework [17], while the Rep-PAN component integrates RepVGG and CSP modules. It also features a Decoupled-Head [22] and Anchor-Free design [23]. YOLOV7 introduces E-ELAN (Extended Efficient Layer Aggregation Networks), which builds on ELAN [24] concepts, and RepConvN based on RepConv [21]. YOLOV9 incorporates PGI (Programmable Gradient Information), consisting of a main branch, auxiliary reversible branch, and multi-level auxiliary information, while GELAN (Generalized Efficient Layer Aggregation Network) combines elements from YOLOV5′s C3 module and YOLOV8′s C2f module. Finally, YOLOV10 employs CIB (Compact Inverted Block).

### 2.3. Training Environment and Evaluation Metrics

The pig ear detection algorithm was trained and tested on a Linux (version 5.8.12) server running a deep learning framework configured on Ubuntu 18.04, with support for CUDA 12.1, Anaconda3, and Python 3.8.18. All experiments were conducted without pretrained weights, setting the number of epochs to 200, batch size to 16, and workers to 8. The input image size was 640 × 640, the optimizer was set to SGD (Stochastic Gradient Descent), and the initial and decay learning rates were both set to 0.01.

The evaluation of the pig ear detection models was performed across six dimensions: loss function, precision (*P*), recall (*R*), mAP50, parameters, and latency, assessing model accuracy and complexity during training, validation, and testing phases. Precision (*P*) reflects the proportion of true pig ear samples among the predicted positive pig ear regions:(1)$$P=\frac{TP}{TP+FP}$$
where *TP* denotes the number of correctly predicted pig ear samples, and *FP* denotes the number of samples incorrectly predicted as pig ears.

Recall (*R*) represents the proportion of correctly predicted pig ear samples among all actual pig ear samples:(2)$$R=\frac{TP}{TP+FN}$$
where *FN* is the number of false negatives, representing missed detection of pig ear samples.

The mean Average Precision (mAP) indicates the average precision (*AP*) over the *P-R* curve area. The mAP can be calculated as follows:(3)$$mAP=\frac{\sum_{i=1}^{N}\int_{0}^{1}P(R)dR}{N}\times 100\%$$
where *N* is the number of classes; here, *N* is 1. In this study, mAP50 represents the mean *AP* at an IoU threshold above 0.5.

The loss function provides an optimization objective for model training. Minimizing this function enhances model prediction accuracy. *Parameters* denote the total number of trainable parameters (such as weights and biases), measuring model complexity. *Latency* measures the time required for a single forward inference, reflecting the model’s inference speed.

### 2.4. Dataset Establishment

This section details how TIRPigEar dataset was established including the collection samples, annotation and labeling information, data storage locations, and data value.

#### 2.4.1. Data Environment

Due to the use of a pig inspection robot capable of flexible movement and adjustable collection height, this dataset includes images taken from various viewpoints, enriching the diversity of the data. Each pig stall contains only one pig, and most images captured by the robot contain a single pig, enhancing the dataset’s utility for individual pig assessment.

Figure 4 shows thermal infrared images of pigs captured in their natural status. These images, focusing primarily on the pig’s head, can support research on temperature distribution of pig heads.

#### 2.4.2. Annotations and Labels

Pig ear data were annotated using the open-source tool *LabelImg*, employing rectangular bounding boxes, and saved in three formats: Pascal VOC, COCO, and YOLO. Each format meets the requirements of various object detection algorithms, with the structure illustrated in Figure 5.

To ensure annotation accuracy and consistency, annotators were trained using standardized guidelines, and a quality control process was implemented, involving periodic reviews and corrections of the annotations. Additionally, a final verification step was conducted by experienced reviewers to ensure the correctness of the annotations.

In Figure 5, the labels in YOLO format are the most concise. Specifically, Pascal VOC format labels provide category names, image names, image paths, and the length, width, and depth information of pig ear labels; COCO format labels provide category names, image names, length and width of pig ear labels, and coordinate information in the image; and YOLO format labels provide category codes, length and width of pig ear labels, and coordinate information in the image.

#### 2.4.3. Data Organization

Thermal infrared pig ear images from the nine data collection rounds are stored in nine separate folders. After data cleaning and annotation, each folder retains between 2000 to 3000 images, with a folder structure as shown in Figure 6. The numerical suffix in each folder name indicates the distribution of image counts across the nine rounds.

The images and corresponding labels in each folder were randomly divided into training, validation, and test sets at a ratio of 8:1:1. Figure 7 shows the structural relationship among pictures, folders, and labels in each training set, validation set, or test set. Each set folder contains Pascal VOC, COCO, and YOLO format labels, making them compatible with different detection tasks. The internal structure of each training, validation, and test set folder is consistent, as illustrated in Figure 7.

#### 2.4.4. Value of the Data

This dataset is valuable for monitoring and analyzing pig state in pig farms, particularly in applications of thermal infrared (TIR) pig ear recognition using deep learning convolutional neural networks.

The thin structure and rich vascular network of the pig ear make it ideal for thermal infrared imaging, as ear temperature data can be quickly extracted to detect potential issues in pigs.

The pig ear’s thermal distribution, combined with information from traditional ear tags or electronic identification, provides insights into individual pig status, aiding farms in early intervention and environmental regulation.

Details about the TIRPigEar dataset are presented in Table 3.

The TIRPigEar dataset also supports using FLIR Tools analysis software for determining pig ear temperatures, which can provide body temperature estimations. The FLIR Tools software (version Flir Tools 201804) for image temperature analysis, image interpretation, and report generation can be downloaded at the direct URL (https://github.com/maweihong/TIRPigEar, accessed on 20 December 2024). Under the ‘Data Download’ section in the README.md file at this link, click on the FLIR Tools download link to obtain the software.

The FLIR IP Config software (version FLIR_IP_Config_3_5) for configuring and diagnosing FLIR network cameras and IP devices is available at the direct URL (https://github.com/maweihong/TIRPigEar, accessed on 20 December 2024). Under the ‘Data Download’ section in the README.md file at this link, click on the FLIR IP Config download link to obtain the software.

The steps for determining pig ear temperature information using FLIR Tools are as follows:

Step 1. Configure the thermal imaging camera using FLIR IP Config software to capture and process thermal infrared images of pigs.

Step 2. Use deep learning algorithms to identify pig ear regions, generating bounding box text data.

Step 3. Import the captured TIR images and pig ear bounding box text data into FLIR Tools.

Step 4. FLIR Tools computes and displays temperature data for the pig ear region, including maximum, minimum, and average temperatures.

Step 5. Use regression analysis to establish a linear or nonlinear model between pig ear temperature and body temperature.

Step 6. Based on the model and pig ear temperature distribution, may estimate the pig’s body temperature [25].

Step 7. Export measurement results as a CSV report for further analysis.

The temperature data of pig ear regions calculated with FLIR Tools are pixel-level temperature values, offering a potential way for higher temperature detection accuracy. Users can use the pictures provided in the TIRPigEar dataset to directly import them into FLIR tools to identify the temperature of each pixel in the picture and predict the overall temperature of pigs. Similarly, users can also detect the target at the pig ear first, then import the detection results into FLIR tools to identify the temperature of the pig ear area and take the pig ear temperature identification results as the overall temperature of the pig. In addition to enabling object detection, the publicly available dataset can also be used for instance segmentation to study temperature distribution. The primary focus of this paper is on pig ear detection within the detected target area.

## 3. Results and Discussion

This section analyzes various object detection algorithms for pig ear detection, highlighting the TIRPigEar dataset’s adaptability. We compare the accuracy, parameter sizes, and latency of models like RT-DETR and YOLO series to assess deployment potential on edge devices. Loss, precision, recall, and mAP50 evaluations follow, with visualizations confirming TIRPigEar’s robustness for real-world applications.

### 3.1. Comparison of Detection Performance of Different Algorithms

Considering the speed and efficiency advantages of single-stage object detection algorithms when deployed on resource-constrained edge devices, this study employs mainstream single-stage object detection algorithms for training and testing the pig ear detection model. By testing various detection algorithms on the same dataset, the superior quality and outstanding performance of the proposed dataset can be demonstrated. The object detection algorithms used in the experiments include the RT-DETR model [26] and the YOLO series of models [27], specifically YOLOv5 [28], YOLOv6 [29], YOLOv7 [30], YOLOv8 [31], YOLOv9 [32], YOLOv10 [33], and YOLOv11 [34].

A total of 2931 thermal infrared images from the TIRPigEar dataset (comprising 2344 training images, 293 validation images, and 294 test images) were selected to validate the detection performance of the provided dataset across different algorithms. Figure 8 illustrates the distribution of the dataset used during the pig ear detection training process, with the coordinates represented as normalized values. It can be observed that the distribution of the center points of the bounding boxes for pig ear targets in the dataset is relatively uniform. Additionally, the widths and heights of the bounding boxes follow a linearly decreasing distribution pattern, indicating a uniformity in the shape characteristics of the pig ear.

As shown in the left part of Figure 9, it is evident that YOLOv7 exhibits relatively lower mean average precision (mAP) compared to other methods across varying parameter counts. The parameter configurations used for the models were selected to evaluate the trade-offs between accuracy, computational efficiency, and latency. For instance, within the range of 10–35 M parameters, YOLOv9 demonstrates superior average precision compared to other algorithms. YOLOv5 is known for its efficient parameter utilization, and it can be observed that variations in parameter counts do not significantly affect its precision, making it a strong choice for resource-constrained environments. When different methods achieve an identical mAP50 of 98%, YOLOv10 has eight times fewer parameters than YOLOv8, illustrating the impact of parameter selection on model complexity and efficiency.

As depicted in the right part of Figure 9, under similar performance conditions, YOLOv9 reduces latency by 28.85% compared to YOLOv5, while YOLOv10 reduces latency by 30.21% compared to RT-DETR. The decision to use these specific parameter configurations was based on the goal of balancing performance and resource usage, enabling an evaluation of the models across a broad spectrum of computational requirements. YOLOv5n, with the smallest number of parameters at 1.76 M, achieves a corresponding mAP50 of 98.2%, while YOLOv11-n, with the least latency at 1.6 ms, maintains a mAP50 of 97.7%.

These results indicate that smaller models (e.g., YOLOv5n) tend to have fewer parameters, which results in faster inference times but can sometimes sacrifice a slight decrease in accuracy compared to larger models. On the other hand, models with more parameters (e.g., YOLOv9, YOLOv10) generally achieve higher accuracy but with increased computational cost and latency. The decision to use these different parameter configurations was to provide a broad understanding of model performance under varying resource constraints.

Table 4 presents the detection performance of various algorithms in the pig ear detection task. As the model size, parameters, and computation decrease, inference speed increases, and resource consumption lessens. Among these models, YOLOv5n, YOLOv8n, YOLOv9t, YOLOv10n, and YOLOv11n demonstrate faster inference speeds with lower resource consumption. In contrast, YOLOv5x, YOLOv8x, and YOLOv9e show poorer performance across these three key metrics. Additionally, models with higher Precision, Recall, and mAP50 scores include YOLOv5n, YOLOv9s, YOLOv9m, and YOLOv9e. Therefore, YOLOv5n may serve as an initial choice for deploying pig ear detection models on mobile or embedded devices.

Figure 10, Figure 11, Figure 12, Figure 13 and Figure 14 illustrate the changes in loss values and precision during the training of the RT-DETR and YOLO series pig ear detection models. In terms of loss curves, the YOLOv9 series models exhibit higher loss values, stabilizing below 15. The models with lower loss values are the YOLOv5 and YOLOv7 series models, where the loss value of YOLOv5 stabilizes below 0.2, and that of YOLOv7 stabilizes below 0.04. The model loss reflects the degree of difference between the predicted results for pig ears and the actual ear labels; a lower loss value indicates better performance in recognizing pig ears. However, excessively low loss values may indicate potential overfitting of the model.

Regarding changes in precision, most models maintain a pig ear detection precision of over 90%. However, the precision of the YOLOv6 series models for pig ear detection falls below 80%, indicating that the YOLOv6 series lacks strong classification capabilities for pig ears. This also suggests that the TIRPigEar dataset may not be suitable for application within the YOLOv6 series models.

Overall, as the number of training epochs increases, the loss values of various algorithms gradually decrease and stabilize, while precision gradually improves and stabilizes. When employing the YOLOv7 model for pig ear recognition, the lowest loss value reached is as low as 0.023. In contrast, when utilizing the YOLOv9m model for pig ear recognition, the highest precision achieved is 97.35%.

The detection performance of each model is comprehensively evaluated using recall and mAP50 (mean Average Precision at Intersection over Union of 0.5), as depicted in Figure 15, Figure 16, Figure 17, Figure 18, Figure 19, Figure 20, Figure 21 and Figure 22. Notably, the recall curve of YOLOv7x exhibits significant fluctuations during the first 50 epochs, with its mAP50 curve also showing multiple substantial variations. This indicates some instability in the performance of the YOLOv7x model during pig ear detection. Conversely, the recall rate of the YOLOv6 series models remains consistently low, ultimately stabilizing at around 80%. The observation of a relatively high mAP50 value in the YOLOv6 series models, combined with the results shown in Figure 11, suggests that these models possess poor generalization capabilities.

In contrast, all models, excluding YOLOv7x and the YOLOv6 series, demonstrate high recall rates and mAP50 values. Among all tested models, YOLOv11s achieves the highest recall rate at 98.5%, while YOLOv11l records the highest mAP50 at 98.7%. YOLOv9m also performs well, with a recall rate of 98.1% and a mAP50 of 98.6%, indicating superior overall performance. The TIRPigEar dataset exhibits high accuracy across most object detection models, showcasing strong capabilities in recognizing pig ear targets. Overall, the TIRPigEar dataset is of high quality and is adaptable to various object detection algorithms.

### 3.2. Visualization Analysis of Pig Ear Detection

To address the interpretability limitations of deep learning algorithms, we performed visualization analysis on inference results of various infrared pig ear detection models using the test dataset. Figure 23, Figure 24, Figure 25, Figure 26, Figure 27, Figure 28, Figure 29 and Figure 30 show the performance of different object detection algorithms on infrared images of pig ears, with results examined under medium-, close-, and long-range image conditions.

Among the detection results across all images, the YOLOv5 series algorithms achieved a maximum confidence score of 0.97 and a minimum of 0.55 in detecting pig ears. The YOLOv6 series had a maximum confidence of 0.96 and a minimum of 0.67. The YOLOv7 series showed a maximum confidence of 0.98 and a minimum of 0.88. For the YOLOv8 series, the maximum and minimum detection confidence scores were 0.98 and 0.61, respectively. The RT-DETR series achieved a maximum confidence of 0.95 and a minimum of 0.45, while the YOLOv9 series had a range from 0.95 to 0.5. The YOLOv10 series showed a maximum confidence of 0.96 and a minimum of 0.44, and the YOLOv11 series displayed a maximum confidence of 0.94 and a minimum of 0.57. These results indicate that all algorithms accurately identified pig ear targets, with most algorithms achieving detection confidence above 0.9.

In medium-range images, the YOLOv8x algorithm achieved the highest pig ear detection confidence at 0.98, while the YOLOv10b algorithm exhibited missed detections. For close-range images, the YOLOv8x algorithm had the highest detection confidence of 0.96, though missed detections were observed with the YOLOv5s, YOLOv9e, YOLOv10b, and YOLOv11n algorithms. In long-range images, the YOLOv7-tiny algorithm achieved the highest detection confidence of 0.98, while the YOLOv6n algorithm exhibited false positives, with a false detection confidence score of 0.67. These findings demonstrate that the TIRPigEar dataset supports high-precision detection performance across various object detection algorithms and includes images with differing detection difficulties. This highlights the diverse advantages of the TIRPigEar dataset, including images captured from varying angles, distances, and environmental conditions. The results confirm the high quality of this dataset, showing that models trained on the TIRPigEar dataset are capable of adapting to the complex field environments of practical livestock farms, facilitating model deployment and application.

## 4. Conclusions

The thermal distribution of pig ears can reflect related issues, and monitoring abnormal ear temperatures is crucial for timely intervention and environmental management in livestock farming. In this study, we developed a new dataset, TIRPigEar, for pig ear detection using thermal infrared imaging technology to address the limitations of existing infrared thermal imaging datasets. The TIRPigEar dataset includes 23,189 thermal infrared images of pig ears, each annotated in Pascal VOC, COCO, and YOLO formats. Through data collection, cleaning, and annotation, this dataset serves as a high-quality foundational resource for future research. Following the dataset construction, pig ear detection experiments were conducted using different object detection algorithms, with results showing the best performance on the YOLOv9m model. The YOLOv9m-based pig ear detection model achieved a maximum accuracy of 97.35%, a recall rate of 98.1%, and a mAP50 of 98.6%. These findings demonstrate the high quality of this thermal infrared pig ear dataset, offering new avenues for swine state monitoring.

## 5. Limitations

This method only provides accuracy data for pig ear detection, and its ability to accurately reflect pig temperature has not reached the ideal level. This limitation may stem from various factors, including environmental influences, constraints of measurement devices, or algorithmic shortcomings. Consequently, further research should focus on improving temperature measurement accuracy to better reflect the actual physiological state of the pigs.

While ear temperature can serve as an indicator of a pig’s status, the correlation between its distribution characteristics and the pig’s overall state requires further investigation. Current studies may not fully account for the physiological mechanisms behind changes in ear temperature. Therefore, developing a more comprehensive state assessment model calls for improvements and extensions to existing research methods.

## Figures and Tables

**Figure 1 animals-15-00041-f001:**
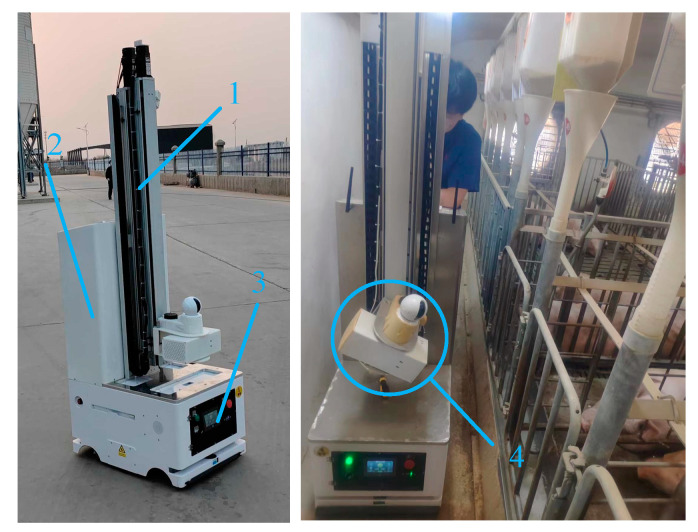
Inspection robot for pig farms. 1: Lifting mechanism for state inspection; 2: information acquisition and control unit; 3: mobile platform; 4: image acquisition component—infrared thermal imager.

**Figure 2 animals-15-00041-f002:**
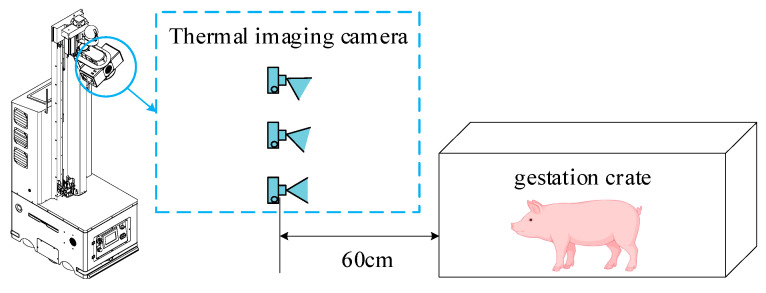
Illustration of thermal infrared imaging for pig ear data collection. The camera’s capture angle was set within 90°, with minor angle adjustments to ensure full coverage of the pigs’ head.

**Figure 3 animals-15-00041-f003:**
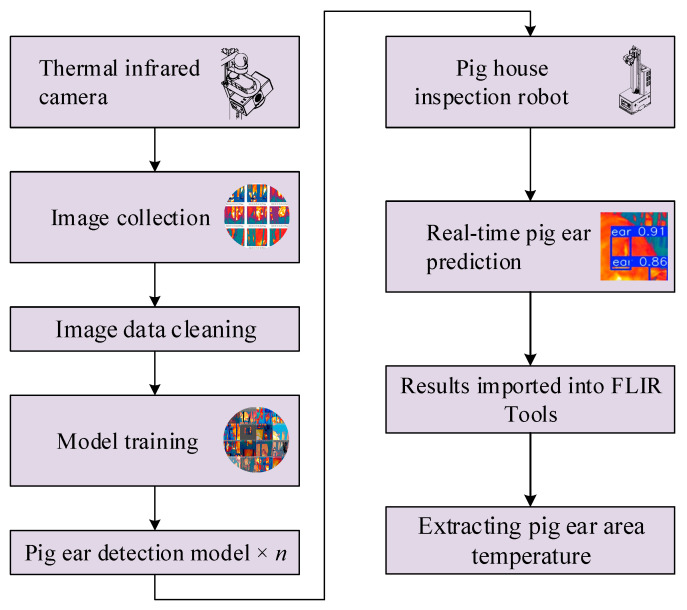
Data collection workflow for the dataset.

**Figure 4 animals-15-00041-f004:**
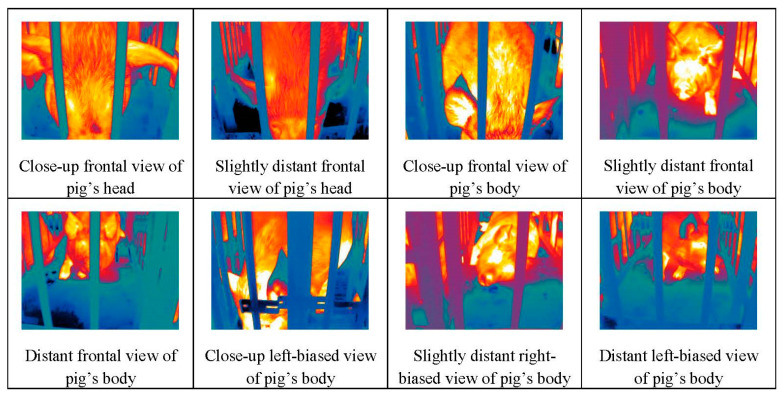
Examples of thermal infrared images of pigs in the dataset.

**Figure 5 animals-15-00041-f005:**
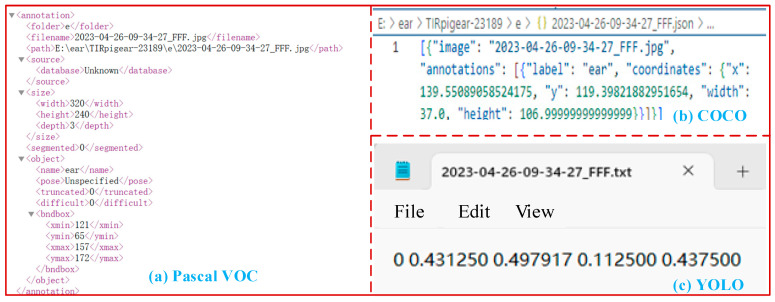
Composition of Pascal VOC, COCO, and YOLO datasets. Pascal VOC labels are saved as .xml files, COCO labels as .json files, and YOLO labels as .txt files.

**Figure 6 animals-15-00041-f006:**
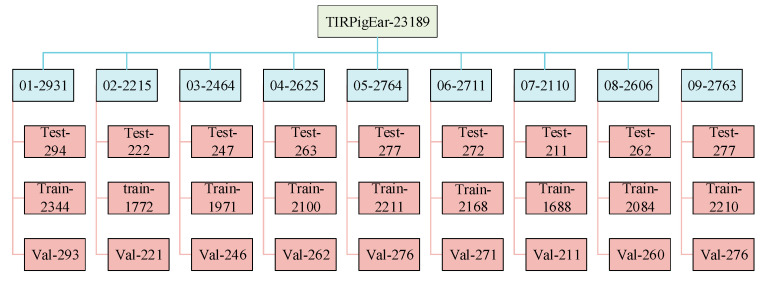
Dataset structure.

**Figure 7 animals-15-00041-f007:**
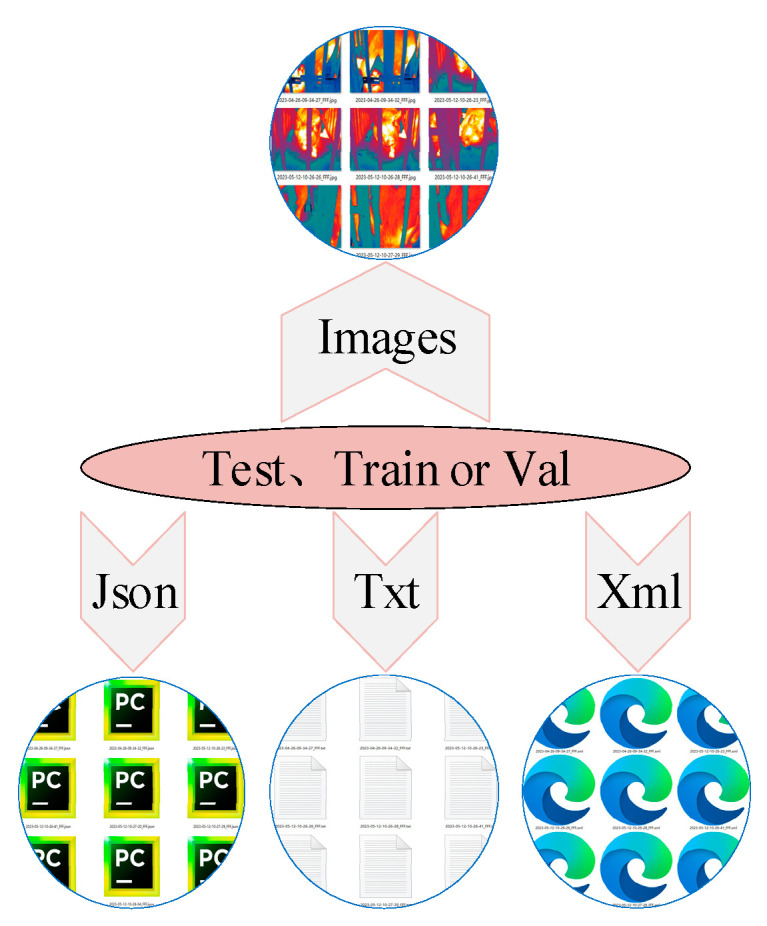
Internal structure of training, validation, and test set folders.

**Figure 8 animals-15-00041-f008:**
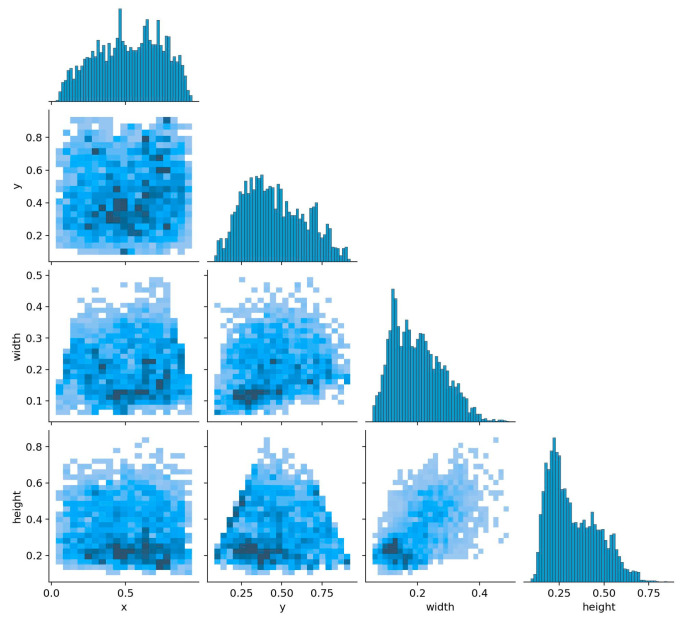
Quality assessment of the pig ear detection dataset. (x, y) represents the center coordinates of the bounding box; width indicates the bounding box width; height represents the bounding box height. The different shades of blue in the figure correspond to the density of the label distribution: deeper shades of blue indicate areas where the labels are more densely concentrated.

**Figure 9 animals-15-00041-f009:**
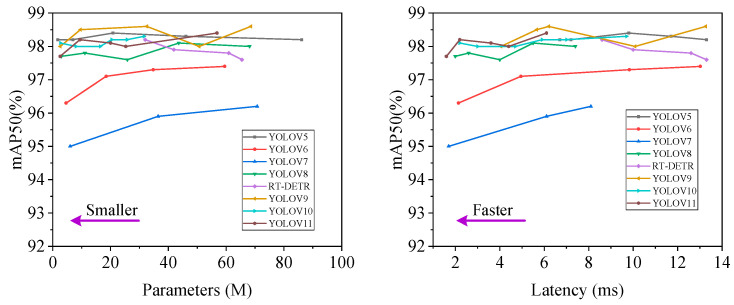
Comparison of accuracy vs. parameter size (**left**) and latency vs. accuracy (**right**) using different methods on the pig ear detection dataset.

**Figure 10 animals-15-00041-f010:**
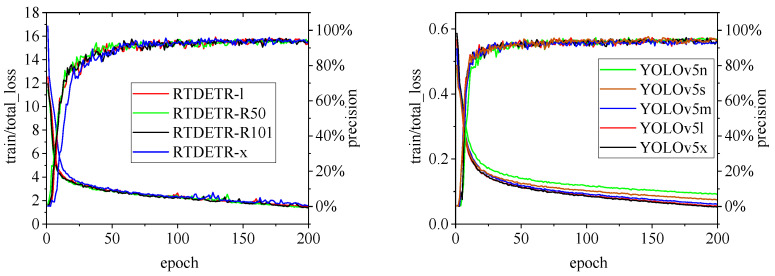
Variations in loss values and precision for the RT-DETR and YOLOv5 pig ear detection models.

**Figure 11 animals-15-00041-f011:**
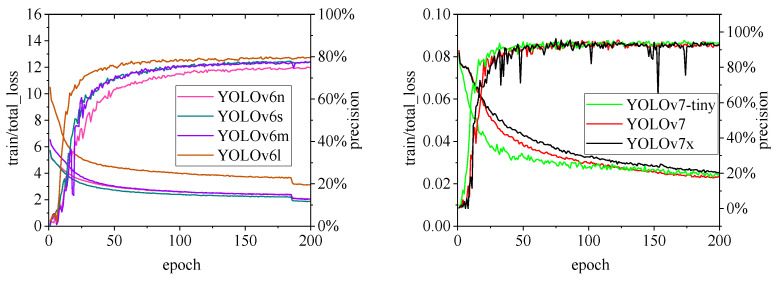
Variations in loss values and precision for the YOLOv6 and YOLOv7 pig ear detection models.

**Figure 12 animals-15-00041-f012:**
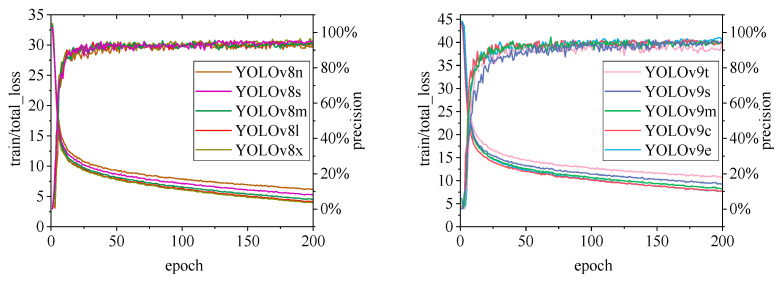
Variations in loss values and precision for the YOLOv8 and YOLOv9 pig ear detection models.

**Figure 13 animals-15-00041-f013:**
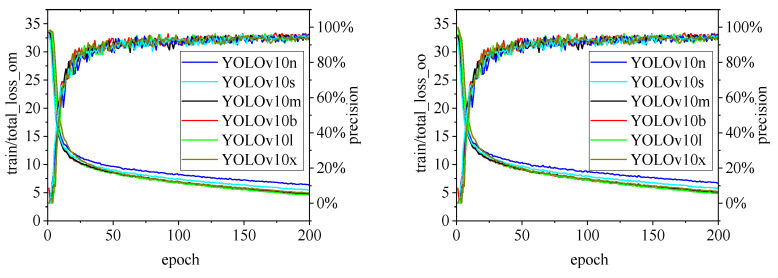
Variations in loss values and precision for the YOLOv10 pig ear detection model.

**Figure 14 animals-15-00041-f014:**
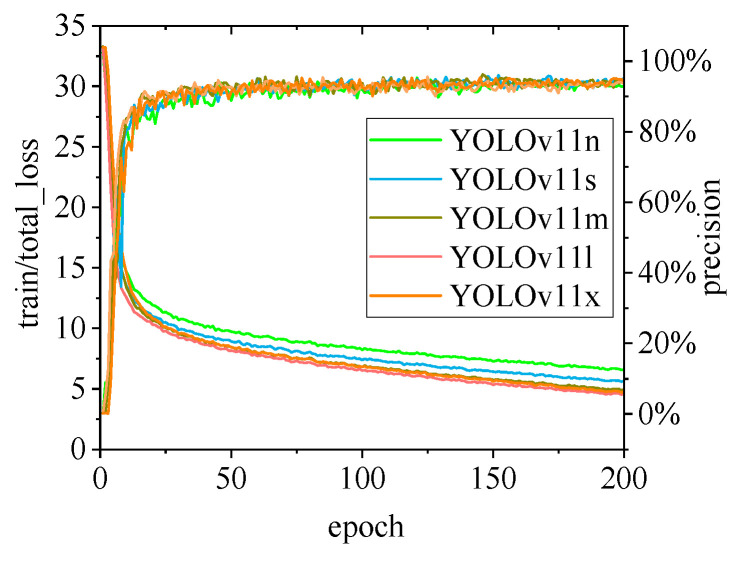
Variations in loss values and precision for the YOLOv11 pig ear detection model.

**Figure 15 animals-15-00041-f015:**
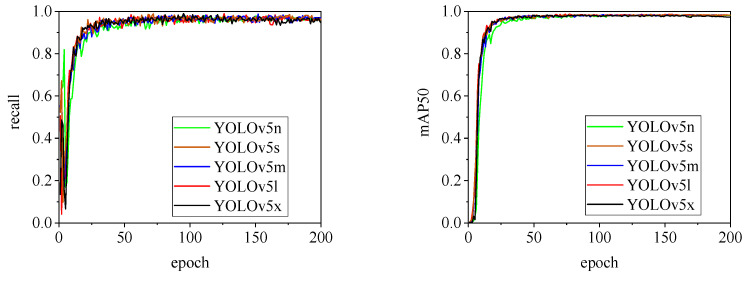
Recall and mAP50 variation curves for the YOLOv5 series models.

**Figure 16 animals-15-00041-f016:**
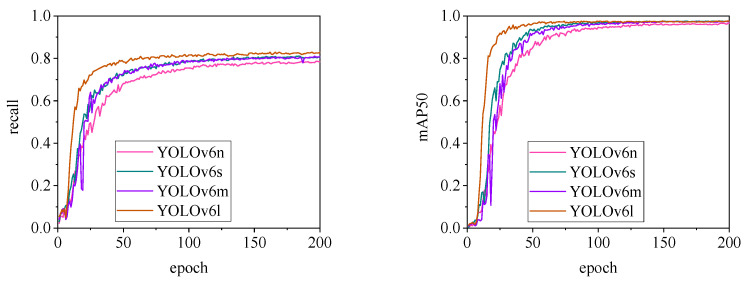
Recall and mAP50 variation curves for the YOLOv6 series models.

**Figure 17 animals-15-00041-f017:**
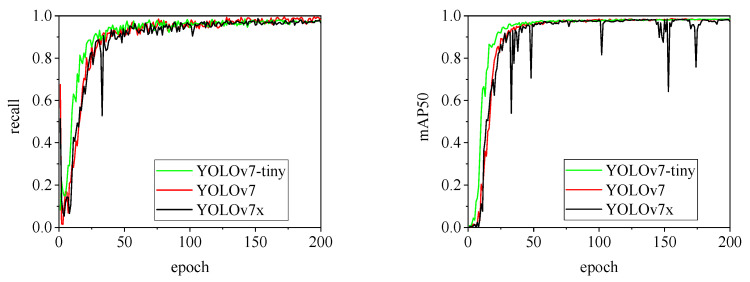
Recall and mAP50 variation curves for the YOLOv7 series models.

**Figure 18 animals-15-00041-f018:**
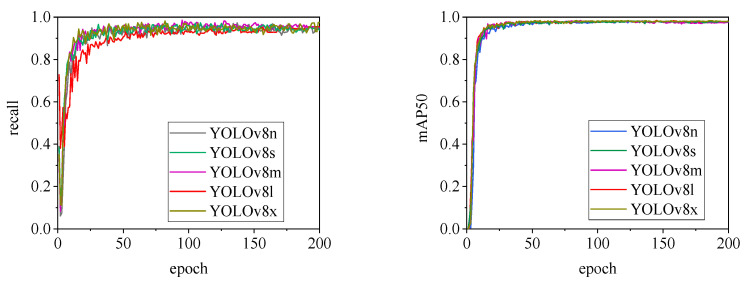
Recall and mAP50 variation curves for the YOLOv8 series models.

**Figure 19 animals-15-00041-f019:**
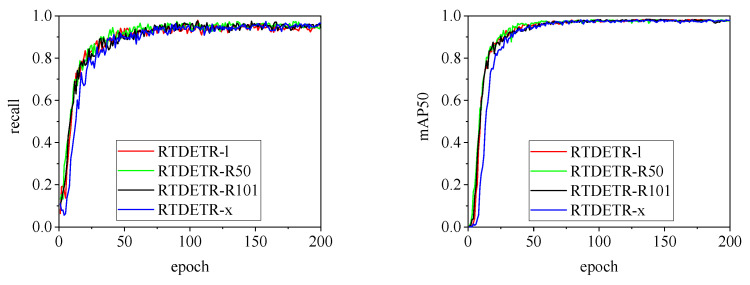
Recall and mAP50 variation curves for the RTDETR series models.

**Figure 20 animals-15-00041-f020:**
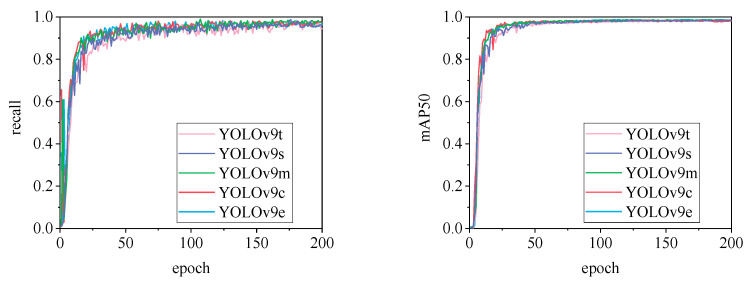
Recall and mAP50 variation curves for the YOLOv9 series models.

**Figure 21 animals-15-00041-f021:**
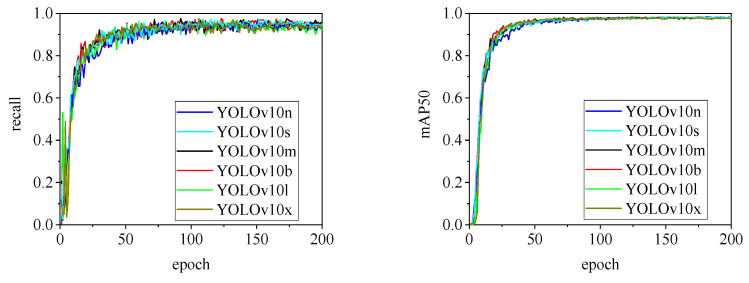
Recall and mAP50 variation curves for the YOLOv10 series models.

**Figure 22 animals-15-00041-f022:**
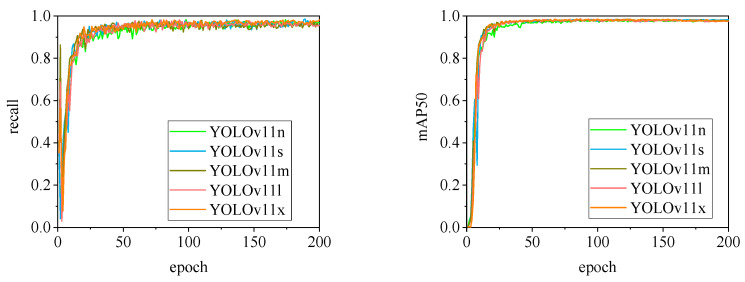
Recall and mAP50 variation curves for the YOLOv11 series models.

**Figure 23 animals-15-00041-f023:**
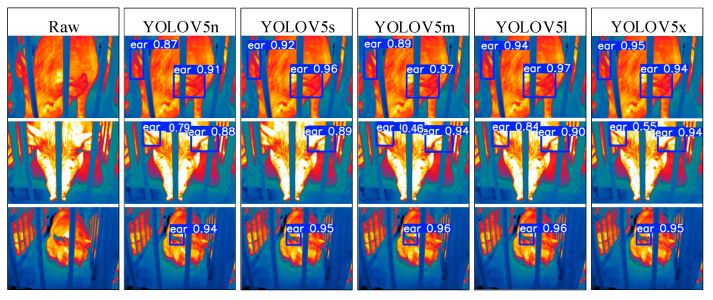
Pig ear detection results of the YOLOv5 series algorithms.

**Figure 24 animals-15-00041-f024:**
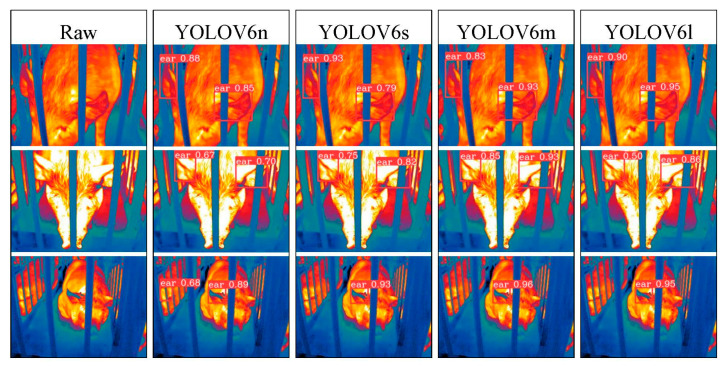
Pig ear detection results of the YOLOv6 series algorithms.

**Figure 25 animals-15-00041-f025:**
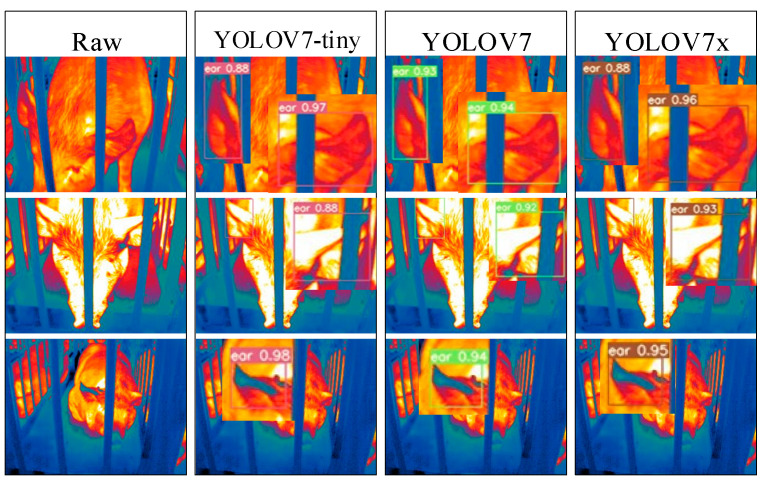
Pig ear detection results of the YOLOv7 series algorithms.

**Figure 26 animals-15-00041-f026:**
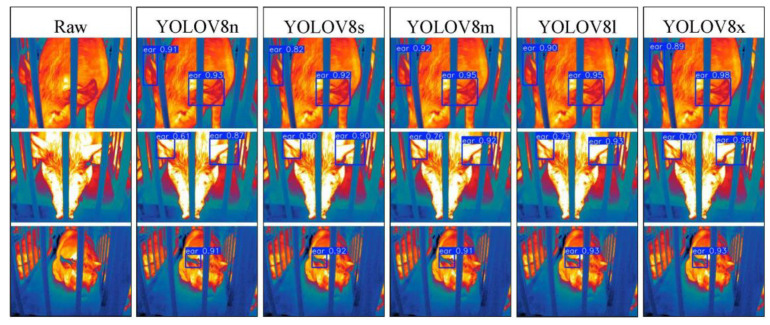
Pig ear detection results of the YOLOv8 series algorithms.

**Figure 27 animals-15-00041-f027:**
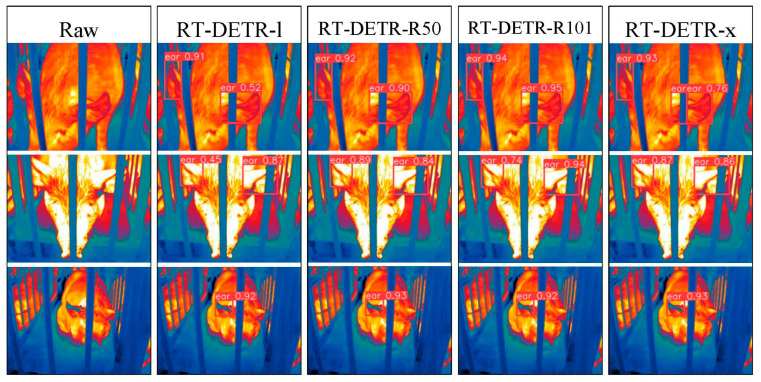
Pig ear detection results of the RT-DETR series algorithms.

**Figure 28 animals-15-00041-f028:**
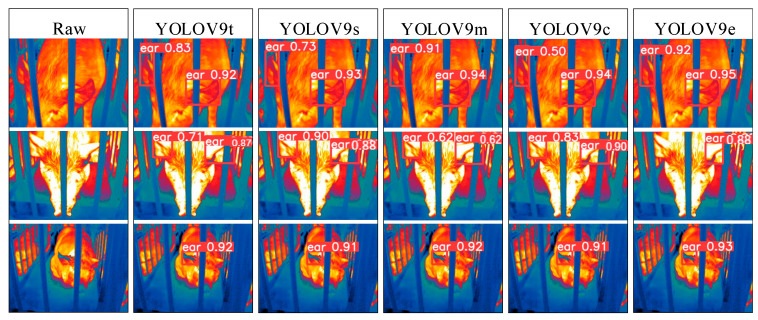
Pig ear detection results of the YOLOv9 series algorithms.

**Figure 29 animals-15-00041-f029:**
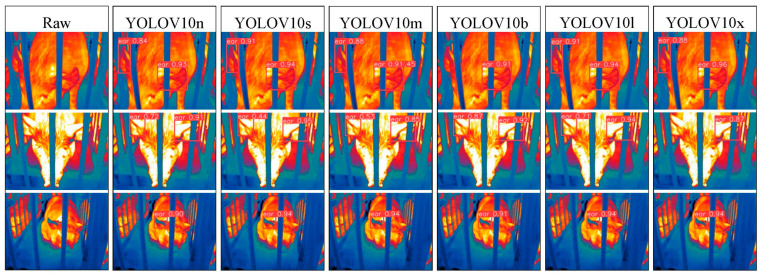
Pig ear detection results of the YOLOv10 series algorithms.

**Figure 30 animals-15-00041-f030:**
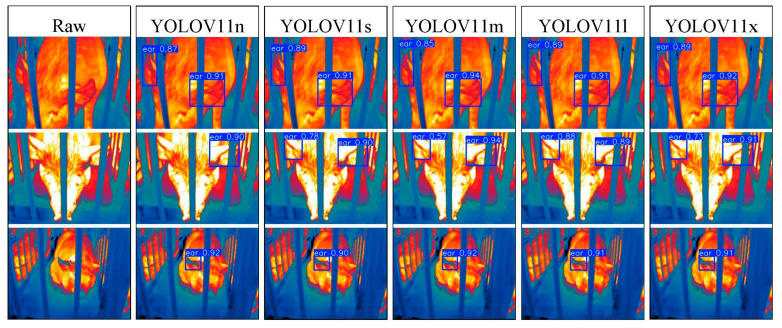
Pig ear detection results of the YOLOv11 series algorithms.

**Table 1 animals-15-00041-t001:** Specifications of experimental apparatus.

Device No.	Category	Description
1 Inspection Robot	Navigation Mode	Manual/Automatic Magnetic Navigation
	Cruise Speed	Ten-level speed adjustment: 3–30 m/min
	Control System	Industrial PC, Windows 10 × 64, WiFi+4 G connectivity
	Lifting Device Dimensions	820 mm × 600 mm × [adjustable height: 514–2519 mm]
2 Infrared Thermal Imager	Camera Manufacturer	FLIR Systems
	Camera Model	FLIR A300 9 Hz
	Resolution	320 × 240 pixels
	Field of View (FOV)	Expanded from 25° to 90°

**Table 2 animals-15-00041-t002:** Comparison of pig ear detection models by series.

Model	Developer	Release Year	Backbone Highlights	Neck Highlights	Head Highlights
YOLOV5	Ultralytics	2020	C3 module, SPPF module	FPN, PAN	Coupled-Head, Anchor-based
YOLOv6	Meituan	2022	RepVGG or CSPStackRep.	Rep-PAN	Decoupled-Head, Anchor-Free
YOLOV7	Chien-Yao Wang et al.	2022	E-ELAN	E-ELAN	Auxiliary head, Anchor-Base
YOLOv8	Ultralytics	2023	C2f	PAFPN	Decoupled-Head, Anchor-Free
RT-DETR	Baidu	2023	Backbone	Hybrid encoder	Decoder
YOLOv9	Chien-Yao Wang et al.	2024	PGI	GELAN	Decoupled-Head, Anchor-free
YOLOv10	Tsinghua University	2024	CIB	CIB	Nms-free, Anchor-free
YOLOv11	Ultralytics	2024	C3k2, C2PSA	C3k2	Decoupled-Head, Anchor-Free

**Table 3 animals-15-00041-t003:** TIRPigEar dataset information.

Subject	Dataset of Thermal Infrared Images of Pigs
Field	Deep learning-based object detection, pig ear recognition, temperature monitoring, state analysis.
Data Format	Thermal infrared images, “.xml” labeled data, “.txt” annotated data, “.json” annotated data.
Data Type	Thermal infrared image.
Data Collection	Thermal infrared images were collected using a pig inspection robot equipped with infrared thermal imaging in a swine farm. The image resolution is 320 × 240, with a total of 23,189 images and a compressed dataset size of 7.71 GB.
Collection Location	Country: China; City: Mianyang, Sichuan; Institution: TQLS Guanghui Core Breeding Farm.
Data Accessibility	Repository: TIRPigEar Direct URL: https://github.com/maweihong/TIRPigEar (accessed on 20 December 2024)

**Table 4 animals-15-00041-t004:** Comparison test results of different models.

Subject	Model Size (MB)	Parameters (M)	Computation (GFLOPs)	Precision (%)	Recall (%)	mAP50 (%)
RT-DETR-l	66.2	32.0	103.4	92.8	94.7	98.2
RT-DETR-ResNet50	86.0	41.9	125.6	93.5	95.5	97.9
RT-DETR-ResNet101	124.2	60.9	186.2	92.6	94.6	97.8
RT-DETR-x	135.4	65.5	222.5	94.0	94.9	97.6
YOLOv5n	3.9	1.8	4.1	95.4	95.9	98.2
YOLOv5s	14.4	7.0	15.8	94.5	96.3	98.2
YOLOv5m	42.2	20.6	47.9	93.3	96.6	98.4
YOLOv5l	92.8	46.1	107.6	94.7	96.6	98.3
YOLOv5x	173.1	86.2	203.8	94.5	96.1	98.2
YOLOv6n	9.95	4.7	11.4	75.0	78.4	96.6
YOLOv6s	38.7	18.5	45.3	78.0	97.5	80.5
YOLOv6m	72.5	34.9	85.8	77.7	97.3	80.8
YOLOv6l	114	59.6	150.7	79.8	97.6	82.8
YOLOv7-tiny	12.3	6.0	13.0	95.8	95.1	95.0
YOLOv7	74.8	36.5	103.2	94.8	95.3	95.9
YOLOv7x	142.1	70.8	188.0	92.9	96.5	96.2
YOLOv8n	6.3	3.0	8.2	92.3	93.6	97.7
YOLOv8s	22.5	11.1	28.6	94.8	95.4	97. 8
YOLOv8m	52.1	25.8	78.9	91.4	96.6	97.6
YOLOv8l	87.7	43.6	165.2	92.3	96.6	98.1
YOLOv8x	136.7	68.1	257.8	91.9	97.5	98.0
YOLOv9t	6.1	2.6	10.7	90.2	96.7	98.0
YOLOv9s	20.3	9.6	38.7	92.2	97.8	98.5
YOLOv9m	66.2	32.6	130.7	92.8	98.1	98.6
YOLOv9c	102.8	50.7	236.6	95.0	96.1	98.0
YOLOv9e	140.0	68.5	240.7	97.0	95.6	98.6
YOLOv10n	5.8	2.7	8.2	93.5	94.6	98.1
YOLOv10s	16.6	8.0	24.4	93.7	95.0	98.0
YOLOv10m	33.5	16.5	63.4	92.4	95.3	98.0
YOLOv10b	41.5	20.4	97.9	94.3	93. 4	98.2
YOLOv10l	52.2	25.7	126.3	93.6	93.7	98.2
YOLOv10x	64.1	31.6	169.8	93.4	95.6	98.3
YOLOv11n	5.5	2.6	6.3	93.8	96.6	97. 7
YOLOv11s	19.2	9.4	21.5	94.6	95.4	98.2
YOLOv11m	40.5	20.0	68.0	94.9	95.3	98.1
YOLOv11l	51.2	25.3	86.6	92.7	96.3	98.0
YOLOv11x	114.4	56.8	194.4	92.6	97.6	98.4

## Data Availability

Direct URL to the data: https://github.com/maweihong/TIRPigEar (accessed on 20 December 2024).

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
