# Peer review of "Leveraging Thermal Infrared Imaging for Pig Ear Detection Research: The TIRPigEar Dataset and Performances of Deep Learning Models"

_animals, 2024, doi:10.3390/ani15010041_

Round 1

Reviewer 1 Report

Comments and Suggestions for Authors

Manuscript Review Animal 3377162 Ma et al.

General Comments

This is a very interesting manuscript suggesting the use of a robot and infrared thermal scan system for monitoring health conditions in pigs. The technical fabrication of the robot and thermal camera are very creative and the authors are to be commended for their efforts. In addition, the testing of various machine learning models is an appropriate component of such an application.

A primary deficiency in the manuscript is the disconnect between the biology of a pig and the technology. The ear of a pig and the role the ear plays as a thermoregulatory organ is very complex. That entire discussion with appropriate references is missing from the manuscript.  Moreover, the manuscript proposes the automated scan system would have utility in the monitoring of animal health. However, there is no data available to demonstrate such health related relationships.

As the manuscript exists, this is a very technical and specific reporting of a mechanical scan system validation and as such the manuscript is perhaps more suited to the technical section of an engineering type journal. The journal Animal is by contrast focused more on the biology of the animal per se. Again, that is not to take away any of the creativity of the authors efforts in developing this scan system. An appropriate approach may be to first publish the scan system and technical validation in an engineering or instrumentation journal and then conduct an appropriate scan on an animal model which included validated health aberrations.  

Specific Comments

Abstract – again, the manuscript is presented as an animal health scanning system but no actual animal health data is presented.

L17-18 and L23-25 -the manuscript suggests the pigs ear is ideal for thermal imagery due to clear temperature readings representing core temperatures. I would respectfully point out that the pig’s ear is thermally very complex. The ear is used to regulate core temperature, not represent it. The ear temperature is quite variable, not thermally symmetrical, prone to “cavity” temperature readings and displays significant variation in time form minute to minute. The authors themselves have actually experienced this (L448  - methods do not yet provide an accurate reflection of body temperature).

Introduction

-the manuscript is devoid of any biological background references regarding thermoregulation in pigs particularly pertaining to animal health (example Berckmans 2022 Advances in precision livestock farming, Burleigh dodds or Luzi et al 2013 Thermography, Brescia and others).

L99 – the use of gestating sows has the added thermoregulatory complexities of heat management of the conceptus.

-There are some 30 figures. This is perhaps necessary for the validation of the engineering models but is quite excessive for a biologically focused journal such as Animal. These could/should be reduced.

Reviewer 2 Report

Comments and Suggestions for Authors

Scientific Article Evaluation Report

1. Summary and Aim of the Study

The aim of the study is clear and well defined, focusing on building a database using infrared thermal images of pigs' ears and evaluating the performance of deep learning models. However, the term “temperature distribution” in the title does not match the results obtained in the study. In addition, the expression “new model” may give the impression that a new model has been created, when in reality the authors are comparing existing models. It is therefore recommended to remove “temperature distribution” from the title, as there are no results related to this topic.

2. 2 Materials and Methods

2.1 Experimental Method

Figure 2: There is a term that appears above the pig in Figure 2 that needs to be translated.

The methodology used to choose the detection models needs more detail:

FLIR Tools: The explanation of the use of FLIR Tools is unclear. The tool was mentioned, but no detailed explanation of its use or information on where to download it was provided. It is recommended to better explain how FLIR Tools works and include the download link.

Choice of Detection Models: The choice of models such as YOLOV5, YOLOV6, YOLOV7, YOLOV8, RT-DETR, YOLOV9, YOLOV10, and the YOLOV11 series needs a more detailed explanation. I suggest explaining how these models were selected and the advantages of each for the study.

Table 2: The meaning of the * below Table 2 is unclear. We recommend explaining what the asterisk (*) represents, so that readers can understand the data correctly. Detail the Yolov model8.

Figure 4: There are duplicate figures with different captions that are not mentioned in the text. The captions should be adjusted to ensure consistency, or the figure should be revised to avoid duplication.

2.4.2 Annotations and Labels and 2.4.3 Data Organization

Figures 5 and 7: Figures 5 and 7 seem unnecessary as they do not contribute significantly to the discussion. I suggest removing them or replacing them with a clearer explanation.

2.4.4 Value of the Data

Segmentation in the Dataset: The use of FLIR Tools allows temperature values to be obtained at pixel level, which offers greater precision for detecting high temperatures. However, no segmentation was used in the labeling of the dataset, which could prevent the inclusion of pixels from other parts of the animal's body. I suggest including an explanation of why segmentation was not applied and what the implications of this are for the results.

Model parameter selection: In Figure 9, there is a parameter selection in the models that is not well explained. It is necessary to clarify how this selection of parameters was made, which would help to understand the impact of this choice on the results obtained.

3. Results and Discussion

3.1 Comparison of Detection Performance of Different Algorithms

The figures illustrating the results can be reorganized to improve clarity and avoid visual overload:

Figures 10 to 14: Figures 10 to 14 present separate results, but together they overload the reading. I recommend combining the results into a single graph, highlighting only the best performance of each model series, which would make it easier to interpret the data.

Figures 15 to 17: Figures 15 to 17 also show excessive results and can be combined into a single graph to reduce redundancy and facilitate analysis.

From topic 4 Conclusion we jump to topic 6 Limitations

6. Limitations

In the limitations section, the authors state that current methods do not provide an ideal level of accuracy in reflecting the body temperature of pigs. However, the metrics obtained to support this claim were not presented. It is recommended that the authors include the metrics of accuracy, error and other relevant metrics to better support this limitation and contextualize the results.

8. Conclusion

The article presents an interesting and relevant proposal, but there is a need for adjustments on several points to ensure greater clarity and robustness in the methodological explanations and results. The above suggestions, if implemented, will certainly increase the quality of the study.

Final Recommendations:

Accept with minor changes

Justification: The corrections mentioned are necessary, but do not compromise the integrity of the article.

Comments on the Quality of English Language

The article needs an English review

Reviewer 3 Report

Comments and Suggestions for Authors

This paper focuses on detecting pig ear information using thermal infrared imaging technology. Constructing the TIRPigEar dataset and experimenting with various deep-learning algorithms, demonstrates certain innovative features and application potential. The paper is well-structured and logically coherent, and its experimental design is reasonable with detailed data analysis. However, several aspects of its exposition and experimental design could be further improved:

1. The annotation process should be described in detail, including measures to ensure annotation accuracy and consistency, such as training methods for annotators and quality control measures.

2. More specific statistical data on the distribution of images under different environmental conditions in the dataset should be provided to help readers better understand the dataset’s diversity.

3. In Table 2 (L156), the semicolons are used incorrectly.

4. The numbering of some figures and tables in the text is not sequential. For instance, Figure 4 appears twice. The numbering format should be checked and standardized.

5. Figure 5 (L227), which presents the composition structure of the Pascal VOC, COCO, and YOLO datasets, suffers from poor overall quality and ineffective information delivery. Improvements are recommended.

6. Table 4 (L320) provides detailed performance data of different models across multiple metrics but lacks clear identification of the best and worst-performing models under each metric, along with comparative analysis. This makes it difficult for readers to quickly and intuitively grasp the performance differences among models. Enhancements are suggested.

Round 2

Reviewer 3 Report

Comments and Suggestions for Authors

All my concerns have been resolved in the revision.